# Spatial changes in park visitation at the onset of the pandemic

**Kelsey Linnell**[1]*, **Mikaela Irene Fudolig**[1], **Aaron Schwartz**[2], **Taylor H. Ricketts**[2], **Jarlath P. M. O'Neil-Dunne**[3], **Peter Sheridan Dodds**[1], **Christopher M. Danforth**[1]

**1** Vermont Complex Systems Center, University of Vermont, Burlington, VT, United States of America, **2** Gund Institute for Environment, University of Vermont, Burlington, VT, United States of America, **3** Spatial Analysis Laboratory, University of Vermont, Burlington, VT, United States of America

* klinnell@uvm.edu

## Abstract

The COVID-19 pandemic disrupted the mobility patterns of a majority of Americans beginning in March 2020. Despite the beneficial, socially distanced activity offered by outdoor recreation, confusing and contradictory public health messaging complicated access to natural spaces. Working with a dataset comprising the locations of roughly 50 million distinct mobile devices in 2019 and 2020, we analyze weekly visitation patterns for 8,135 parks across the United States. Using Bayesian inference, we identify regions that experienced a substantial change in visitation in the first few weeks of the pandemic. We find that regions that did not exhibit a change were likely to have smaller populations, and to have voted more republican than democrat in the 2020 elections. Our study contributes to a growing body of literature using passive observations to explore who benefits from access to nature.

## Introduction

Parks are important public infrastructure that provide a venue for interaction with nature, socialization, and exercise. Park access and use has been found to offer both mental and physical health benefits [1–6]. Among the many benefits of exposure to nature are faster healing, decreased stress and increased ability to manage life's challenges [7, 8]. During the COVID–19 pandemic, access to parks may have been important for mitigating and managing the secondary impacts of the virus. Recent publications indicate that access to parks during the pandemic is important for a variety of reasons including providing a venue for exercise, increasing happiness, and improving social cohesion [9–14].

While park visitation may have provided significant support to personal and public health at the time, it is unclear whether park visitation changed, to what extent, and for whom in the United States. In March of 2020, stay at home orders were issued in most states, and many non-essential workplaces and public spaces were closed. Following these events, overall mobility decreased dramatically for most Americans, reaching a maximum reduction by 34 to 69% depending on the state [15, 16]. While Americans were visiting fewer locations in general, some research suggests that park visitation may not have been subject to this decline. An early study of parks on the West Coast determined changes in visitation at the onset of the pandemic to be primarily motivated by seasonal change, while a study of parks in New Jersey found that

**Data Availability Statement:** Data is available for purchase from UberMedia (now Near). That the authors did not receive special access to this data beyond that which any other researcher may

potentially have access to. Near can be contacted through their website at: https://near.com/contact/.

**Funding:** KL, CD, and PSD were supported by a gift from MassMutual. MassMutual did not participate in the design of the study, or the collection, analysis, or interpretation of data.

**Competing interests:** The authors have declared that no competing interests exist.

early pandemic visitation was higher than the baseline [17, 18]. Together these results indicate that visits to parks may have differed from other points of interest at the onset of the pandemic.

Preliminary examination of trends suggest that changes in park visitation were not universal. In the United States, partisanship, even at the regional level, is associated with behavioral differences. Researchers have found that Thanksgiving dinners were 30 to 50 minutes shorter when the guests and hosts resided in voting precincts that had been in opposition in 2016 [19]. Mobility studies of Americans during the pandemic have found differences along partisan lines as well. The American political system is largely dominated by two political parties: Democrats, and Republicans. This divide in political ideology has been found to be indicative of differing identities and behaviors. This is particularly true of COVID-19 policy response and preferences [20]. Republicans have been found to have lower vaccination rates, have a smaller decrease in mobility during the pandemic, and to be less compliant with non-pharmaceutical interventions [15, 21–25]. Counties with more Republicans also had less severe mobility restrictions, and were less responsive to their governor's recommendation to stay home [15, 26]. Given these partisan differences in general mobility, we seek to determine whether changes in local park visitation at the onset of the pandemic also differed by partisanship, or whether park visitation uniquely transcended these differences.

Studies of park usage in March and April of 2020 have thus far relied on survey data, or have been geographically limited, and neglected to establish a baseline of seasonality of park usage [17, 18, 27, 28]. Here we utilize mobile device data from across the United States to explore abrupt non-seasonal changes in park visitation at the regional level. We use data from 2019 to discern seasonal visitation patterns, and employ a change-point detection algorithm to diagnose sudden changes in behavior at the onset of the pandemic. By classifying regions by whether or not an abrupt change in park visitation took place, we are able to discern whether or not these abrupt changes occurred along partisan lines. We conduct further comparisons across population, income, and share of employment by industry to provide insight into other factors that may have influenced whether or not an abrupt change occurred. In the Data section we introduce the data used to classify regions, and make these comparisons. The Methods section then gives a detailed explanation of the data aggregation for each region, and the classification procedure and methods of comparison applied to the aggregated data. The results of the comparisons are described, and then discussed.

## Data

To determine whether a partisan effect is observed in park visitation we used park visitation data from across the United States, and voting share data from the 2020 Presidential Election. Differences in regions with and without abrupt changes in visitation were further analyzed using population estimates, and income and employment share data from the US Census and the Bureau of Economic Affairs. Details for these data sources are provided below.

### Park visitation data

Our park visitation dataset was acquired from UberMedia (now part of Near), and consists of daily visitation counts for non-commercial parks for each day of 2019 and 2020. There are 8,135 parks in the data set, including municipal, neighborhood, and city parks. National and State Parks were specifically excluded as predominantly travel destinations. Parks are located in each of the 50 states, and Washington DC. A total of 1,033 counties, roughly a third of all counties, contain at least one park from our dataset.

Daily visitation counts were determined using location data from mobile devices. Each unique device appearing within a park's bounds on a single day was counted as a visit. A device's location was reported when an individual used one of over 400 apps utilizing a GPS Software Development Kit (SDK) in partnership with UberMedia (90% of data by volume), or when a user interacted with an advertisement through real time bidding on one of over 250,000 apps (10% of data by volume). GPS location and an accompanying timestamp were determined from the device's operating system.

The number of devices reporting activity in at least one location in the US on a given day is referred to as the Daily Active Users (DAUs). This number refers to all locations, not simply parks. In 2019 and 2020 the monthly DAUs varied between 38 and 60 million, and represented roughly 10% of the adult population in the United States.

In mid December 2019 the set of SDK's in partnership with UberMedia was updated. This change in data collection corresponded to a large increase in observations throughout the US, and was not spatially uniform. Thus, while the raw 2019 and 2020 park visitation data are not directly comparable, we analyze their relationship where possible.

### Voting and economic data

Voting data at the state and county levels from the 2020 election was retrieved from MIT Election Data and Science Lab, and is available at https://electionlab.mit.edu/data.

The BEA publishes data on employment by industry (using the North American Industry Classification System (NAICS)) for each county in table "CAEMP25N", which can be found at https://apps.bea.gov/regional/downloadzip.cfm. In this table "Farming" and "Forestry" are considered as separate, though they appear as one sector in the NAICS classification. For this study they were considered separately, as they appear in the table. County level population, income, and economic data from the 2019 American Community Survey were obtained from US census API.

### Methods

In this study we compare regions in the United States where abrupt non-seasonal changes in park visitation did and did not occur at the onset of the pandemic. In order to make these comparisons we begin by classifying regions as having or not having an abrupt change point by applying a change point detection algorithm to aggregated time series of park visitation in each of the regions (Fig 1). Once regions have been classified, the distributions of the regions across population, income, share of employment by industry, and voteshare in the 2020 presidential election are compared.

### Aggregation

Daily visits to a park were defined as the unique number of mobile devices reporting GPS coordinates found inside the park polygon bound on a day. This daily visit count was then normalized by the average Daily Active Users for the month in which it was found, approximating the percentage of devices observed in parks relative to all observed devices. The normalized visitation was then summed over each week in order to minimize noise. Weekly visitation was summed for parks contained in a county, or a state, and thus a time series of weekly park visitation between 2019 and 2020 was created for each county and state containing at least one park from our data set.

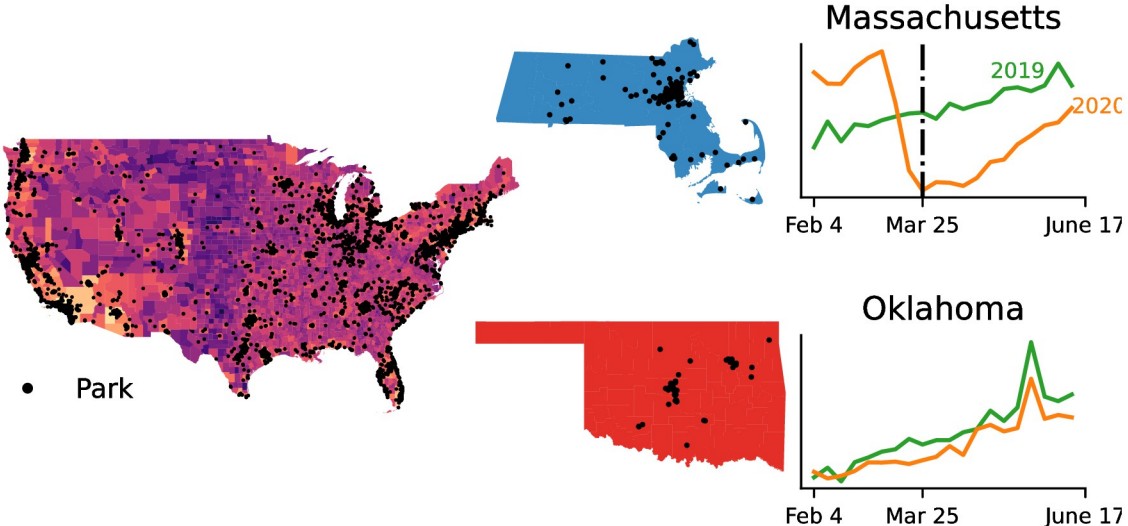

**Fig 1. A heat map of the population of the contiguous United States in log scale overlaid with the locations of the parks used in the study, with each park demarcated with a black point.** Observation of this map indicates that the parks in this data set have an urban bias, and that the parks are roughly distributed according to population distribution in the United States. Maps of park locations are shown for Massachusetts (top) and Oklahoma (bottom), with the color of each map representing the political party receiving the most votes in the 2020 Presidential election (Democrats for Massachusetts, and Republicans in Oklahoma). Normalized weekly park visitation for each state is plotted to the right. Visitation for 2019 is plotted in blue, while visitation for 2020 is plotted in orange. For Massachusetts there is a significant dip in visitation bottoming out the week of March 25, 2020, and the visitation plots for 2019 and 2020 diverge. For Oklahoma visitation does not drop off in March, and does not diverge from 2019 visitation patterns. Generated with GeoPandas using Census data [29].

## Change point detection

To determine whether a substantive change in visitation is observed in each time series, we use the Bayesian Estimator of Abrupt Change, Seasonality, and Trend (BEAST) [30]. This method decomposes a time series into a seasonal (harmonic) component, and trend (linear) component, and uses Bayesian Inference to fit a model which estimates the location of change points in either of the components. BEAST was chosen because the underlying model acknowledges the seasonal nature of most park visitation time series (more visits in summer). By specifying a 52 week season length, we were able to train the model to the annual cycle shape of the data.

Parametric methods applied without the seasonal decomposition are susceptible to under estimating change points in these particular time series because of the combination of seasonality and the proximity in the series of the data collection change in December 2019 to the onset of the pandemic in March 2020. The initial event represents a sharp increase in visitation volume (roughly 150 pct), while the second appears, for most regions, as a sharp decline. When fit with a single model, these two features appear together as a change in variance, and a parametric model can be nicely fit using a single change point in December 2019.

By decomposing the time series and forcing a decoupling of the two events by specification of seasonal, length we make each event visible as a unique discontinuity in the linear component.

The December 2019 discontinuity could then be accommodated with a trend change point, which incorporates a discontinuity into the linear component. In this way the model was fit while accounting for seasonality, and the abrupt change in data volume.

Allowing a trend change point to be used as described above, the model was effectively limited to selecting a single trend change point, which enabled it to identify the most likely change

point in the data. It is possible for the algorithm to detect no change point, reducing concern that one would be identified artificially.

Regions which had a change point occurring in between mid March and mid April 2020 were considered to have had an abrupt change in park visitation coinciding with the onset of the pandemic and social distancing measures. If a region was found not to have had a change point in this window, it can be assumed that either no change point was found in the time series, or any change occurring in the specified window was not as significant as a change at another time.

Changes induced by seasonality are in most cases more gradual than those that occur in the window of interest, and these changes are accounted for by the harmonic component of the model. The harmonic component is fit using both 2019 and 2020 data, which informs the model of the expected seasonal shape. Since these changes are accounted for in the model fitting, it is unlikely that change points identified in the window of interest are due only to seasonal variation. Because the length of the time series only included two seasons (park visitation demonstrates a yearly cycle), it was not pertinent to search for changes in the seasonal structure.

BEAST is less effective in identifying change points in time series with high variance. The recorded park visits in some of the counties were low enough that the behavior of only a few individuals could have large impacts on the time series itself. To ensure that BEAST was only considering counties for which there was enough data we used a mean normalized visitation threshold of $10^{-5.5}$ (this corresponds to about 120 visits per week in the month with the least DAUs) in 2020. A total of 322 counties did not meet this criteria and were excluded from further analysis. The remaining 711 counties that contain parks in our dataset met this criteria. The counties included in the analysis are roughly 21% of all the counties in the United States, and span all of the states. Details on the selection of the visitation threshold can be found in the Supplementary Materials (See S1 Fig).

## Comparison

Regions were binned according to whether a change point in mid March 2020 was identified or not, and comparisons of the populations of the regions in each category were made. Using data from the 2020 election, states and counties were assigned a percent of the population having voted either Republican (Trump and Pence) or Democrat (Biden and Harris) in the 2020 election. Counties were assigned personal incomes, and population counts using Census estimates from 2019. Voting records and census data were combined to determine the votes cast per capita for each county. Finally, a fraction of employment (employment share) for each industry in the North American Industry Classification System (NAICS) was assigned to each county in the study using data from the BEA. Counties which had no available employment data for an industry were exclcuded from the analysis of that particular industry. Counties with and without detected change points were compared across vote share, population, votes per capita, personal income, and industry employment using Kolomogorov-Smirnov two sample tests. This test was chosen for its ubiquity in the literature and ability to compare distributions with different sample sizes. The means of the distributions are compared using Welch's t-test, which also accommodates different sample sizes.

## Results

### Partisanship in abrupt changes

With the parameters discussed in the Methods section, BEAST found a change point in the window of interest for 21 states, while the remaining 29 states did not exhibit an abrupt change

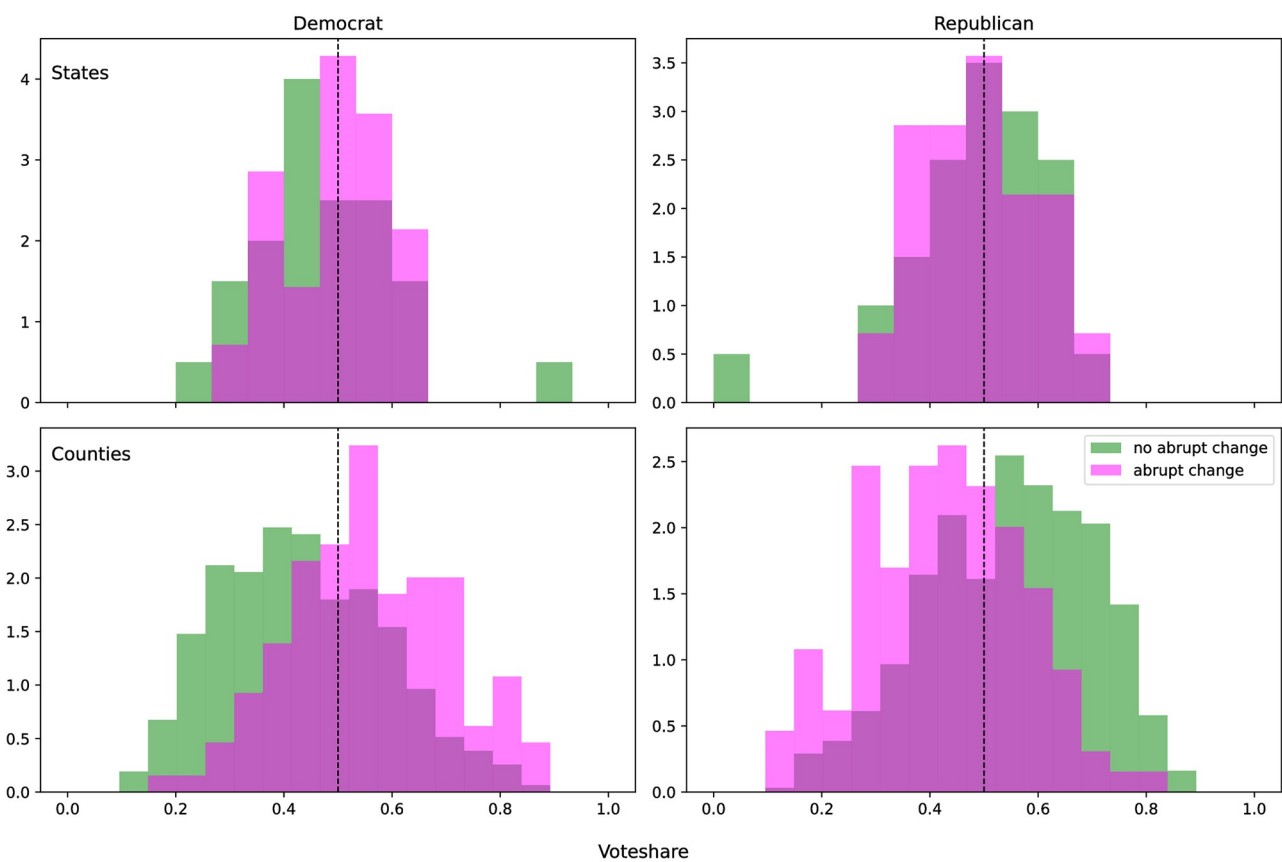

**Fig 2. Top: Distributions of states where a pandemic response change point was detected (pink), and not (green), across proportion of votes cast for the Democratic (left) and Republican (right) parties in the 2020 Presidential Election**. The distributions are not significantly different across voting proportion for either party. The apparent outlier in each figure is Washington DC, which did not exhibit a change point, and for which more than 80% of the votes cast were for the Democratic party. Exclusion of DC from the analysis did not change the results. **Bottom: Distributions of counties with (pink) and without(green) detected change points across percent voting for the Democratic (left) and Republican (right) parties in the 2020 Presidential Election**. Distributions across the percent of votes cast for the Democratic party were determined to be significantly different by the Kolomogorov Smirnov 2 sample test (KS statistic = 0.33, $p$ = 4.23e-10). The distribution of counties with a change point was shifted to the right of those without a change point, indicating counties with change points had greater proportions of votes for the Democratic party. The bulk of the mass of the two distributions lies on either side of 0.5, meaning that the majority of counties with a change point are majority Democrat counties. The distributions across percent voting for the Republican party are likewise significantly different (KS statistic = 0.33, $p$ = 2.35e-10), and indicate counties without change points had greater proportions of votes for the Republican party, and were more likely to be a majority Republican county.

in visitation. Comparison of the 2020 presidential election results for states where visitation did and did not change abruptly is shown in the top row of Fig 2. The distributions across vote share for the two sets of states were not significantly different for either the Democratic or Republican parties (KS statistic = 0.2, $p$ = 0.63 and KS statistic = 0.2, $p$ = 0.63 respectively). The distributions for each party are neither similar, nor mirrored. The difference is accounted for by third party votes, most notably Libertarian votes.

Comparison of the distributions across percent voting Libertarian (which accounted for less than 3% of the vote in all states) indicates that the distributions were significantly different (KS statistic = 0.44, $p$ = 0.015), where Libertarians had greater vote share in states without an abrupt change. S2 Fig demonstrates the relative proportion of Democrat, Republican, and third party votes for each state and county. The state appearing as an outlier in the distributions, where Democrats had the highest vote share, and which did not have an abrupt change, is Washington DC, which was treated as a state for this study. Exclusion of Washington DC does not change the results.

Partitioning the data by county led to significantly different distributions across vote share (bottom row of Fig 2). When the BEAST classification procedure was applied to county level aggregations of visitation data, 123 of the 711 counties had abrupt visitation changes at the onset of the pandemic. The distribution across Democratic vote share for counties with abrupt changes is shifted to the right of the distribution for counties without- indicating that Democrats were more likely to have greater vote share in counties with abrupt changes. Kolomogorov Smirnov 2 sample results confirm that these distributions are significantly different (KS statistic = 0.33, $p$ = 4.23e-10). Observation of the same distributions across Republican vote share reveals that Republicans were more likely to have greater vote share in counties without abrupt changes. KS 2 sample test results support that these distributions are also significantly different (KS statistic = 0.33, $p$ = 2.35e-10).

Not only are the distributions significantly different, but across the vote share for each party they are translated across the x = 0.5 line (drawn in red). This line represents the dividing point in the majority party support in a county. This reveals that Democrats were not only more likely to have greater vote share in counties with abrupt changes, they were more likely to hold a majority in those counties. Likewise, Republicans were more likely to hold a majority in counties without abrupt changes.

The distributions of the counties across vote share for the Democratic and Republican parties are not mirrored on account of votes going to third parties, meaning that "not Democrat" is not the same as "Republican". Both distributions taken together support that there is a partisan divide between counties with and without abrupt changes in park visitation at the onset of the pandemic. This is further supported by no significant difference found in the distributions of the counties over percent voting Libertarian (KS statistic = 0.13, $p$ = 0.087).

## Population, income, and employment

Partitioning the data by county allowed further analysis using population, employment, and income data. Differences in distribution across population size, and votes cast per resident, for the counties with and without abrupt changes, are displayed in Fig 3. Counties with an abrupt change had more than twice the mean population of counties exhibiting no change, and fewer votes per resident than counties that did not. The distributions across each of these variables is significantly different (KS statistic = 0.28, $p$ = 1.08e-07 for log 10 scale population, and KS statistic = 0.13, $p$ = 0.045 for votes cast per resident).

The incomes of the counties were not significantly different (KS statistic = 0.10, $p$ = 0.23), as seen in the distributions in Fig 4.

Counties were also compared on the basis of percent employment in each of the 20 NAICS sectors. The distributions of the counties with and without change points across percent of employment were significantly different ($p < 0.05$) for 14 of the sectors. This includes both Farming Employment, and Forestry, Fishing and Related Activities, which comprise a single sector in the NCAIS, but are considered separately here. Of the 14 sectors with significantly different distributions, Welch's T-Tests found only 10 had significantly different means. The distributions for these 10 sectors is shown for counties with and without abrupt changes in Fig 5. For each sector, the box plot to the left shows the distribution over the fraction of employment for counties with an abrupt change (pink), and the box plot to the right represents the same distribution for counties without an abrupt change (green).

For the 10 sectors with significantly different distributions and means, 5 had higher mean employment share in counties with abrupt changes: Information, Finance and insurance, Professional, scientific, and technical services, Educational services, and Health care and social assistance. These sectors are primarily comprised of white collar workers, and with the

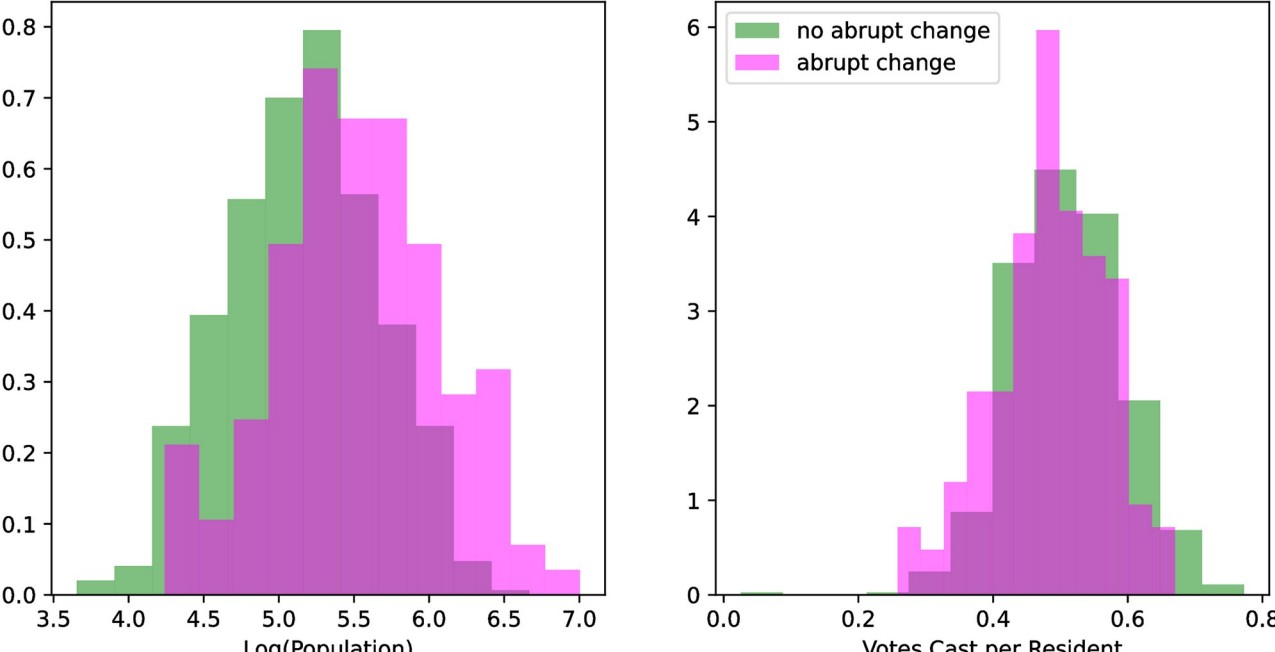

**Fig 3. Distributions of counties with and without change points across the log base 10 of 2019 county population (left) and the votes cast per capita in the 2020 Presidential Election (right).** The distributions over population are roughly log normal, and visibly and significantly different (KS statistic = 0.28, $p$ = 1.08e-07). The mean population of counties with change points was 331,131 people, which is more than twice the mean population of counties without change points, namely 144,544. The counties with the lowest populations were exclusively without change points, while the counties with the greatest populations were exclusively those with change points. The distributions across votes per capita are also visibly and significantly different (KS statistic = 0.13, $p$ = 0.045) with the counties with change points having fewer votes per capita than counties without.

exception of Health care and social assistance, require less onsite work. Farm employment, Mining, quarrying, and oil and gas extraction, Construction, Manufacturing, and Retail trade all had higher mean employment share in counties where abrupt changes in park visitation did not occur.

## Discussion

At the state level, there was no significant difference in the partisanship of regions where an abrupt change in park visitation took place, and those where it had not. There was a significant difference in the vote share of Libertarians, with Libertarians having smaller vote share in states with an abrupt change. However, Libertarian voters account for less than 3% of voters in each state, and are unlikely to be themselves pivotal in deciding overall park visitation behavior for a state. Thus, the practical significance of the difference in Libertarian vote share is doubtful. However, at the county level there is a clear divide in the partisanship of regions where park visitation did and did not undergo abrupt change. Counties with an abrupt change were more likely to be majority Democratic, while counties without a change point were more likely to be Republican. Taken together with the urban bias of the data set, it is possible that the state results are confounded by an over representation of urban park visits.

If abrupt park visitation changes were more associated with Democrat behavior, since urban areas have a Democratic bias, it is possible that the behavior of the urban park goers (who are more likely to be Democrats) may have overshadowed park going behavior in the rural parts of states. This possibility is made further plausible by the observation that the counties with a change point tend to be more populated. If park visitation changes are more likely

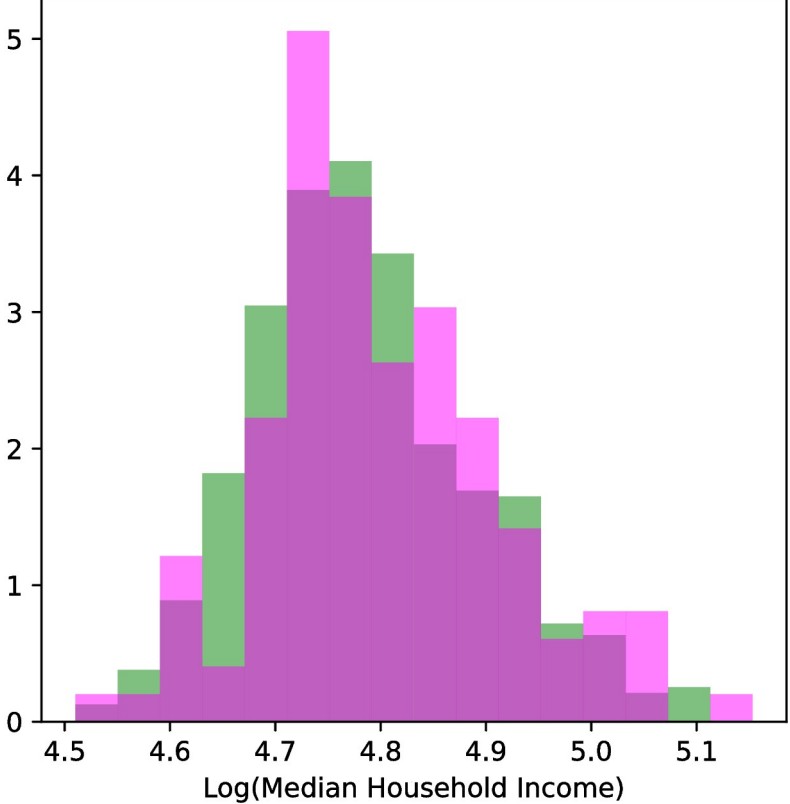

**Fig 4. Distributions of counties with and without change points across the log base 10 of 2019 personal income as reported by the census.** The distributions are visually similar, and not significantly different (statistic = 0.10, p-value = 0.23). There is not a statistically significant difference between the incomes of counties where abrupt park visitation changes occurred and those where it did not.

in areas of greater population, and these areas are also over represented in the data, it stands to reason that aggregation to the state level may obscure behavior of the rural residents in the park visitation data.

Of course there is a second implication of these observations which is that whether or not park visitation exhibited an abrupt change is directly related to population density. If true, this relationship would explain why there is a disparity in population size for counties with and without abrupt changes, and why the counties with the lowest populations did not have abrupt changes, while the counties with the greatest populations did. In this case, differences in party affiliation of the respective areas is possibly unrelated, and only appears due to the confounding correlation between population density and party affiliation [31]. Since there is a connection between small populations and extreme partisanship as well, this would offer a potential explanation for why the span of the distribution across vote share for either party is greater for counties without abrupt changes.

Counties without abrupt changes in park visitation were more likely to have higher proportions of employment in Manufacturing, Construction, Mining, and Farming. Many of the workers in these sectors would have been considered "essential," and much of the work would be site specific. Meanwhile, counties with abrupt changes were more likely to have greater proportions of jobs in Information, Finance and insurance, Professional, scientific, and technical services, and Educational services; sectors where remote work would have been more widely adopted. It is curious that regions with greater proportions of remote workers, who may have had greater

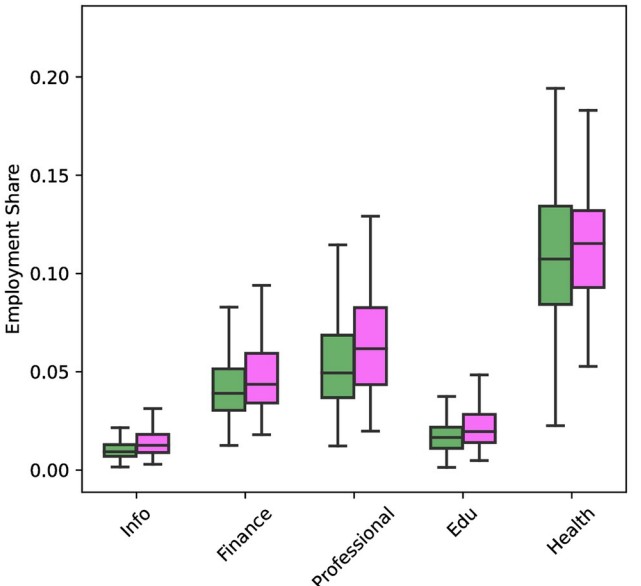
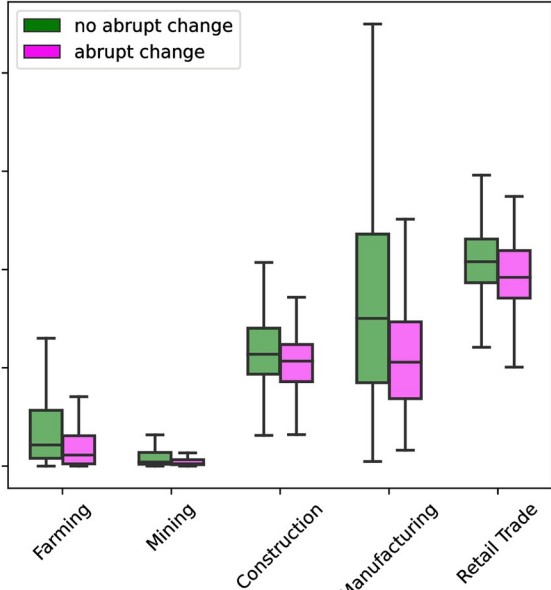

**Fig 5. Box plots showing the distribution across employment share for counties with and without change points in the sectors where the distributions and their means were significantly different.** Sectors in the plot to the left were those where counties with a change point had significantly higher means ($p < 0.05$)), sectors in the plot to the right had significantly greater mean employment share in counties without change points. The distributions across employment share for counties with change points are shown in pink, while the distributions for counties without change points is shown in green. While the differences in mean and distribution for all shown sectors are significant, they are small.

time and opportunity to visit parks at the time, were more likely to experience a drop-off in visits. The difference is interesting and suggests it is possible that reductions in employment related mobility impacted other mobility decisions, such as whether or not to visit parks.

However, while there are differences in employment share by sector, they are small, and their practical significance remains undetermined. The most striking differences found in this study were in population, and partisanship. Recent work [15, 21, 26] suggests that regions with higher Republican vote share exhibited less social distancing at the onset of the pandemic, were slower to adopt stay at home orders, and residents visited more points of interest than residents of regions with higher Democratic vote share, suggesting that overall mobility reduction was greater for Democratic counties than Republican ones. Insight from these new studies suggests that the lack of change in park visitation behavior among Republican regions simply reflects this partisan difference in mobility, and indicates that parks were not necessarily uniquely visited more or less relative to other points of interest.

## Limitations and future directions

This study did not account for differences in local COVID-19 response policies. Incorporation of these differences would be necessary to understand how local governance impacted park access, and how willing residents were to defy local mobility restrictions for parks as opposed to other locations.

The spatial distribution of the parks in our data set roughly corresponds to the spatial distribution of the population, creating a substantial urban bias that we do not control for in this study. Weighing park visitation in such a way to allow for aggregation to the state level without over representing the urban parks would enable more revealing analysis at the state level, and additional insight into the demographic differences between counties with and without change points, with less influence from population density.

Augmenting the current data set with visitation data for more rural parks could also aid in these goals. Greater representation of rural parks would also allow a better investigation into population and park access as it relates specifically to population density and general nature accessibility.

Due to a change in collection methodology at the end of 2019, which led to a spatially non-uniform increase in total visitation counts, we were unable to directly compare 2019 and 2020 data. While there are visibly dramatic dips in behavior for some states and counties at the end of March 2020, it is not possible to clearly quantify how these changes deviate from expected behavior, nor how the magnitude of these changes compare across regions. Future work could investigate other park visitation data, and attempt to use it to normalize and perhaps compare visitation changes. The park visitation estimates used in this study are necessarily underestimates because they rely on mobile devices, and do not account for multiple visits made by a single device in one day. Other measurements of visitation could improve accuracy of visitation estimates.

Comparison of visitation levels across years and regions, especially following the initial pandemic reaction, would be extremely helpful in determining whether or not there were differences in how park visitation was valued in different regions. This could also be achieved by comparing dips in park visitation to dips in visitation to other points of interest. In particular, it would be useful to understand how different areas, and different populations, weigh the benefits and risks of park usage in the pandemic, and how park usage diverted visitation to other destinations. Future work in this direction could benefit from spatio-temporal methods, which would account for any spatial correlation of park visits. Studies indicating which populations had access to parks, which may have been greatly beneficial during 2020, could be used to address potential social inequality, and reduce public health risk in the future.

## Supporting information

**S1 Fig. Plots of the effect of the mean visitation threshold on study results.** *Top*: The mean percent having voted Democrat(left) and Republican (right) in the 2020 Presidential election of the counties with and without change points as the threshold is increased at the log 10 scale. When the threshold is between -8 and -6 the gap in mean vote share between counties with and without abrupt park visitation changes is stable. As the threshold increases past -6 the gap begins to shrink, with the counties with abrupt changes becoming slightly more democrat, and the counties without abrupt changes becoming much more democrat, and both becoming less Republican. *Bottom Left*: The p-value (blue) and statistic(black dashed) results of the KS 2 sample test on the partisan differences in counties with and without abrupt changes as the threshold increases. The pvalue is stable until the threshold is greater than -5, when it begins to increase, but never crosses the $p = 0.05$ significance threshold (red dashed). The k statistic remains stable until the threshold is increased past -6, when it decreases, but never falls below 0.2. *Bottom Right*: The number of counties (black) which meet inclusion criteria as the visitation threshold is increased. There is is rapid decline in counties included in the study beginning at a threshold of -6. Past a threshold of -5 fewer than half of all counties in our data set meet inclusion criteria, and at -4 there are almost none. The number of counties with an abrupt visitation change (pink) remains constant in the study until a threshold greater than -5, reflecting that these are among the counties with the greatest visitation. The number of counties without a change (green) declines almost in parallel to the total (black) counties, indicating that the threshold criteria eliminates these counties almost exclusively.
(TIF)

**S2 Fig. Scatter plots where each state and county is represented by a dot, the color of which corresponds to whether or not an abrupt chagne took place.** The location in the x-y plane is determined by the percent of votes for the Republican(x) and Democratic(y) candidates in the 2020 Presidential Election.
(TIF)

## Author Contributions

**Conceptualization:** Kelsey Linnell, Mikaela Irene Fudolig, Peter Sheridan Dodds, Christopher M. Danforth.

**Data curation:** Kelsey Linnell.

**Formal analysis:** Kelsey Linnell, Mikaela Irene Fudolig, Christopher M. Danforth.

**Funding acquisition:** Christopher M. Danforth.

**Investigation:** Kelsey Linnell, Christopher M. Danforth.

**Methodology:** Kelsey Linnell, Mikaela Irene Fudolig, Christopher M. Danforth.

**Project administration:** Peter Sheridan Dodds, Christopher M. Danforth.

**Software:** Kelsey Linnell.

**Supervision:** Mikaela Irene Fudolig, Peter Sheridan Dodds, Christopher M. Danforth.

**Visualization:** Kelsey Linnell.

**Writing – original draft:** Kelsey Linnell.

**Writing – review & editing:** Kelsey Linnell, Mikaela Irene Fudolig, Aaron Schwartz, Taylor H. Ricketts, Jarlath P. M. O'Neil-Dunne, Peter Sheridan Dodds, Christopher M. Danforth.

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
