## [Decision Letter · Decision Letter 0]

30 May 2022

PGPH-D-22-00198

Spatial changes in park visitation at the onset of the pandemic

Dear Dr. Linell,

Thank you for submitting your manuscript to PLOS Global Public Health. After careful consideration, we feel that it has merit but does not fully meet PLOS Global Public Health’s publication criteria as it currently stands. Therefore, we invite you to submit a revised version of the manuscript that addresses the points raised during the review process, 

We look forward to receiving your revised manuscript.

Kind regards,

Roopa Shivashankar, MD, MSc

Academic Editor

Journal Requirements:

1. Please include details in Funding Information section in the system and ensure that it matches with the Financial Disclosure Statement.

2. Please note that your Data Availability Statement is currently missing a direct link to access each database. If your manuscript is accepted for publication, you will be asked to provide these details on a very short timeline. We therefore suggest that you provide this information now, though we will not hold up the peer review process if you are unable.

3. We ask that a manuscript source file is provided at Revision. Please upload your manuscript file as a .doc, .docx, .rtf

4. Please provide separate figure files in .tif or .eps format and remove from the manuscript file.

5. All figures and supporting information files will be published under the Creative Commons Attribution License (creativecommons.org/licenses/by/4.0/). Authors retain ownership of the copyright for their article and are responsible for third-party content used in the article. 

Figure 1: please (a) provide a direct link to the base layer of the map used and ensure this is also included in the figure legend; (b) provide a link to the terms of use / license information for the base layer. We cannot publish proprietary or copyrighted maps (e.g. Google Maps, Mapquest) and the terms of use for your map base layer must be compatible with our CC-BY 4.0 license. 

Please upload any written confirmation as an 'Other' file type. It must clarify that the copyright holder understands and agrees to the terms of the CC BY 4.0 license; general permission forms that do not specify permission to publish under the CC BY 4.0 will not be accepted. Note that uploading an email confirmation is acceptable.

6. We notice that your supplementary [figures/tables] are included in the manuscript file. Please remove them and upload them with the file type 'Supporting Information'. Please ensure that each Supporting Information file has a legend listed in the manuscript after the references list.

Additional Editor Comments (if provided):

Reviewers' comments:

Reviewer's Responses to Questions

**Comments to the Author**

1. Does this manuscript meet PLOS Global Public Health’s publication criteria? Is the manuscript technically sound, and do the data support the conclusions? The manuscript must describe methodologically and ethically rigorous research with conclusions that are appropriately drawn based on the data presented.

Reviewer #1: Yes

Reviewer #2: Yes

Reviewer #3: Yes

2. Has the statistical analysis been performed appropriately and rigorously?

Reviewer #1: I don't know

Reviewer #2: Yes

Reviewer #3: No

3. Have the authors made all data underlying the findings in their manuscript fully available (please refer to the Data Availability Statement at the start of the manuscript PDF file)?

Reviewer #1: Yes

Reviewer #2: Yes

Reviewer #3: Yes

4. Is the manuscript presented in an intelligible fashion and written in standard English?

Reviewer #1: Yes

Reviewer #2: Yes

Reviewer #3: Yes

5. Review Comments to the Author

Reviewer #1: Summary:

The authors us mobile data (specifically the use of apps or access to advertisements that are part of a database) to estimate weekly park visitation across the United States. They use this data to compare changes in park visitation as a result of the pandemic (2019 vs 2020), and perform secondary analysis to demonstrate differences in park visitation behavior across partisanship, population, income, employment, etc at the level of the state and county.

This is an interesting paper and provides evidence of behavior changes in response to the pandemic. Some of the results are in agreement with respect to expected behavior along axes that were examined (At the county level, Democrats showed more abrupt reductions in park visitations compared to the Republicans).

The authors also highlight the several limitations that arise from the use of secondary data as the basis of their results, which is commendable.

Minor comments are listed below:

The structure of the article seems to be more along the lines of those submitted to engineering and computer science conference proceedings (section I, II etc). Is this format in line with the journal?

Park visitation estimates is likely underestimated (since it solely relies on mobile data) - this is not reported as a limitation

Fig. 1: Visitation for 2019 is plotted in blue - correct to Visitation for 2019 is plotted in green.

In the discussion section, it would be useful to comment on how the abrupt changes in park visitation in certain sections could have affected their health and well-being. In fact, for a journal that focuses on health, it would be ideal to include some health outcomes as well in their analysis with respect to changes in park visitation.

With regards to the statistical analysis, I feel I do not have the necessary technical expertise to evaluate the methods. Given the strong dependence of the results of this article on the advanced statistical methods used, I would recommend that this paper is reviewed by a statistician as well.

Overall, would recommend for publication if the statistical methods are found to be adequate.

Reviewer #2: Author has address the the question well defined manner.

He has explained all section very well. Author has defined the methods properly, shared the data link. From my side there is no comment. We should go ahead with the publication.

Reviewer #3: To summarize, this study presents a fairly comprehensive analysis of data relating to spatial and temporal changes in park visitations, aggregated at different spatial levels. The manuscript is well written in general. The comments below are from the perspective of statistical rigor of the article.

1) While BEAST is chosen as the main analysis vehicle, the authors rely on the KS test to reveal critical findings. This is somewhat a jarring mix and match between frequentist and Bayesian methods. At the very least, this should be acknowledged and clarified in the methods, that after time series analysis and decoupling, the authors chose simpler empirical methods (KS test) for comparisons.

2) The authors rely on a fair amount of manual wrangling and judgement calls to choose the parameters of the Beast application - in terms of profile and number of change points and other settings. Was robustness examined in terms of other settings, such as say, 2 change points?

3) Details of the Beast application (length and number of MCMC chains chosen), as well as characteristics of the MCMC convergence should be commented upon.

4) Spatial correlation may confound some of the findings - since the time series reported and not "independent" sequences of observations. This should at least be acknowledged in the findings and future work should consider spatio-temporal methods.

6. PLOS authors have the option to publish the peer review history of their article (what does this mean?). If published, this will include your full peer review and any attached files.

**Do you want your identity to be public for this peer review?** For information about this choice, including consent withdrawal, please see our Privacy Policy.

Reviewer #1: No

Reviewer #2: **Yes: **Dr. Kalpana Singh

Reviewer #3: No

---

## [Decision Letter · Decision Letter 1]

26 Jul 2022

Spatial changes in park visitation at the onset of the pandemic

PGPH-D-22-00198R1

Dear Dr.Linnell,

We are pleased to inform you that your manuscript 'Spatial changes in park visitation at the onset of the pandemic' has been provisionally accepted for publication in PLOS Global Public Health.

Best regards,

Roopa Shivashankar, MD, MSc

Academic Editor

Reviewer Comments (if any, and for reference):

Reviewer's Responses to Questions

**Comments to the Author**

1. If the authors have adequately addressed your comments raised in a previous round of review and you feel that this manuscript is now acceptable for publication, you may indicate that here to bypass the “Comments to the Author” section, enter your conflict of interest statement in the “Confidential to Editor” section, and submit your "Accept" recommendation.

Reviewer #1: All comments have been addressed

Reviewer #3: All comments have been addressed

2. Does this manuscript meet PLOS Global Public Health’s publication criteria? Is the manuscript technically sound, and do the data support the conclusions? The manuscript must describe methodologically and ethically rigorous research with conclusions that are appropriately drawn based on the data presented.

Reviewer #1: Yes

Reviewer #3: Yes

3. Has the statistical analysis been performed appropriately and rigorously?

Reviewer #1: Yes

Reviewer #3: Yes

4. Have the authors made all data underlying the findings in their manuscript fully available (please refer to the Data Availability Statement at the start of the manuscript PDF file)?

Reviewer #1: Yes

Reviewer #3: Yes

5. Is the manuscript presented in an intelligible fashion and written in standard English?

Reviewer #1: Yes

Reviewer #3: Yes

6. Review Comments to the Author

Reviewer #1: "Visitation for 2019 is plotted in blue, while visitation for 2020 is plotted in orange" ... the authors were requested to change this to

"Visitation for 2019 is plotted in green, while visitation for 2020 is plotted in orange."

This change is yet to be made.

Reviewer #3: The review comments have been addressed satisfactorily.

7. PLOS authors have the option to publish the peer review history of their article (what does this mean?). If published, this will include your full peer review and any attached files.

**Do you want your identity to be public for this peer review?** For information about this choice, including consent withdrawal, please see our Privacy Policy.

Reviewer #1: No

Reviewer #3: No
